# Tracing the Trophic Plasticity of the Coral–Dinoflagellate Symbiosis Using Amino Acid Compound-Specific Stable Isotope Analysis

**DOI:** 10.3390/microorganisms9010182

**Published:** 2021-01-16

**Authors:** Christine Ferrier-Pagès, Stephane Martinez, Renaud Grover, Jonathan Cybulski, Eli Shemesh, Dan Tchernov

**Affiliations:** 1Coral Ecophysiology Team, Centre Scientifique de Monaco, 8 Quai Antoine 1er, MC-98000 Monaco, Monaco; rgrover@centrescientifique.mc; 2Morris Kahn Marine Research Station, Department of Marine Biology, Leon H. Charney School of Marine Sciences, University of Haifa, Haifa 3498838, Israel; stephane.martinez@gmail.com (S.M.); eshemesh@univ.haifa.ac.il (E.S.); dtchernov@univ.haifa.ac.il (D.T.); 3The Swire Institute of Marine Science, The University of Hong Kong, Cape D’Aguilar Road, Shek O, Hong Kong, China; cybulski@connect.hku.hk

**Keywords:** coral, dinoflagellate, Symbiodiniacea, compound specific isotope analysis of amino acids, heterotrophy

## Abstract

The association between corals and photosynthetic dinoflagellates is one of the most well-known nutritional symbioses, but nowadays it is threatened by global changes. Nutritional exchanges are critical to understanding the performance of this symbiosis under stress conditions. Here, compound-specific δ^15^N and δ^13^C values of amino acids (δ^15^N_AA_ and δ^13^C_AA_) were assessed in autotrophic, mixotrophic and heterotrophic holobionts as diagnostic tools to follow nutritional interactions between the partners. Contrary to what was expected, heterotrophy was mainly traced through the δ^15^N of the symbiont’s amino acids (AAs), suggesting that symbionts directly profit from host heterotrophy. The trophic index (TP) ranged from 1.1 to 2.3 from autotrophic to heterotrophic symbionts. In addition, changes in TP across conditions were more significant in the symbionts than in the host. The similar δ^13^C-AAs signatures of host and symbionts further suggests that symbiont-derived photosynthates are the main source of carbon for AAs synthesis. Symbionts, therefore, appear to be a key component in the AAs biosynthetic pathways, and might, via this obligatory function, play an essential role in the capacity of corals to withstand environmental stress. These novel findings highlight important aspects of the nutritional exchanges in the coral–dinoflagellates symbiosis. In addition, they feature δ^15^N_AA_ as a useful tool for studies regarding the nutritional exchanges within the coral–symbiodiniaceae symbiosis.

## 1. Introduction

Marine and terrestrial ecosystems around the world are increasingly threatened by human activities such as climate change, pollution and habitat loss, among others [1,2]. In particular, marine coastal ecosystems are exposed to growing loads of nutrients discharged from the land, which alter the nutrient cycles and stoichiometric balance [3]. In turn, these changes significantly affect the acquisition of nutrients by marine species [4,5]. However, the ability to obtain nutrients in various forms is known to influence animal growth, behavior and reproduction capacity [6], and is one of the key parameters that explain the health status of many animals and their capacity to offset environmental stress [7]. The ability to diversify one’s nutrition source is the reason why mixotrophic organisms, which can obtain their energy from a range of hetero- and autotrophic modes, can adapt to a wide range of environmental situations and are some of the most widespread organisms on Earth [8]. This balance of nutrients as an important biological component and an overpowering pollutant is a dichotomy that many marine animals currently face.

Scleractinian corals are among the most ecologically important mixotrophs that currently face this dichotomy, and are the foundational species that create the most biodiverse marine ecosystem, coral reefs [9]. For energy acquisition, corals rely both on autotrophy through the photosynthetic activity of their endosymbiotic dinoflagellates (the Symbiodiniaceae family, [10]), as well as on heterotrophy/osmotrophy, through predation of plankton by the animal polyps or the uptake of dissolved organic molecules [11,12]. They can also acquire nutrients from the activity of their associated microbes such as diazotrophic bacteria [13]. Taken together, the different partners form a unique entity called the coral holobiont. Autotrophy represents a large percentage of the total energy supply of corals thriving in shallow, clear, oligotrophic waters, while heterotrophy and microbial-mediated nutrition can be important for corals living in deeper or more turbid environments or those affected by a stress event that decreases the autotrophic energy input [14,15]. Overall, heterotrophically sourced food can represent between 30% and 100% of the coral’s energetic needs depending on the environmental conditions [12]. However, heterotrophy is difficult to trace in situ, since corals rely on multiple nutritional sources simultaneously, which are shared amongst the holobiont partners. Therefore, we still lack a basic understanding of the daily contribution of each nutritional source to the energetic budget of corals.

Stable carbon (δ^13^C) and nitrogen (δ^15^N) isotope analyses of animal tissue have been used for over three decades to study dietary preferences in complex environmental contexts (reviewed in [16]). However, the use of bulk stable isotope compositions presents significant limitations in the dietary assessment of animals such as corals, which have complex resources that includes nutrient recycling within the holobiont [17]. Recent coral studies have thus implemented compound-specific isotope analysis (CSIA), using mainly the δ^13^C values of fatty acids and sterols [13,18,19,20]. However, amino acids (AA) contain both carbon and nitrogen atoms on which CSIA can be applied. Although used only twice to study coral diet [21,22], CSIA-AAs have proven to be useful in paleo-studies of the tropical marine nitrogen cycle and for tracing carbon flow in reefs [23,24]. The δ^15^N values of consumers’ AAs give an indication of their trophic level [25,26] because the “source” amino acids (e.g., phenylalanine) fractionate ^15^N minimally with each trophic level, while the “trophic” amino acids (e.g., glutamic acid) are markedly enriched in ^15^N [27,28]. In turn, δ^13^C values can reveal consumers’ diets [29] by tracing essential AAs (AA_ess_), which only plants and microorganisms can synthesize with their own fingerprint. In corals, Fox et al. [21] suggested that δ^13^C-AA_ess_ are useful to trace coral heterotrophy, because they observed a complete separation in the δ^13^C-AA_ess_ values between coral endosymbionts (autotrophy) and the heterotrophic nutritional sources. Fujii et al. [22] used the δ^15^N_AA_ values of several coral species sampled in natural conditions to show that their trophic levels ranged from purely autotrophic (0.71) to partially heterotrophic (1.53). Overall, these two studies show that CSIA-AAs are a promising tool for tracing coral nutritional conditions.

As with early bulk δ^13^C and δ^15^N studies, the validity and efficacy of the CSIA-AA technique still require validation as an efficient tool for tracing nutritional interactions in the coral–dinoflagellate symbiosis. Until now, there has been a very limited knowledge on the synthesis pathways and exchange fluxes within the coral holobiont. It has been shown that the coral host is able to synthesize some or all nonessential AAs [30], mainly by using symbiont-derived glucose [31]. They can also acquire both nonessential and essential AAs through heterotrophy [11,21]. Some works have also highlighted the capacity of the coral hosts to synthesize few AA_ess_ [32,33,34,35] (Figure 1), although the contribution of microbes in such synthesis remains to be investigated. In turn, algal symbionts and bacteria can synthesize all AAs (including AA_ess_, [33]), and transfer them to the coral host for their own use [34]—although how this transfer occurs and in what quantity is yet to be understood. The AA synthesis becomes more complicated when different coral species are analyzed; for example, *Acropora* corals are reliant on their photosymbionts for cysteine production [30], whereas *Porites* corals can synthesize it on its own [33]. Overall, the above studies point out the complexity of the AA synthesis question for the holobiont, indicating its ability to acquire all needed AAs in some capacity (Figure 1).

This pioneering study aimed to validate, using a controlled experimental design, the application of CSIA-AA to study the nutritional ecology of a symbiotic coral–dinoflagellate association and its role in AA acquisition and synthesis. We experimentally produced three sets of corals maintained for several weeks under three nutritional regimes (autotrophy, heterotrophy and mixotrophy) and measured their δ^15^N and δ^13^C-AA signatures. The results obtained clarified the advantages and limitations of the technique as a tool to study coral trophic ecology. Contrary to general belief, we discovered that heterotrophy was best traced in the symbiont rather than in the host fraction, and that the heterotrophic input in mixotrophic corals can vary considerably depending on colonies and energy allocation. This work significantly advances our fundamental knowledge related to the nutritional exchanges between partners of the coral symbiosis. It also expands our understanding of appropriate tools for tracing the nutritional ecology of corals, which is a key prerequisite for understanding food web structure in reefs and ecological dynamics.

## 2. Material and Methods

### 2.1. Experimental Setup

The experiment was performed with individuals of the scleractinian coral *Stylophora pistillata* originating from the Red Sea. Three large nubbins were sampled from six genetically different mother colonies (*n* = 18). Nubbins were then identified, hung on nylon wires, and equally distributed in 3 different nutritional treatments (heterotrophy, autotrophy and mixotrophy), with 3 aquaria per treatment (six nubbins per treatment from different mother colonies, distributed in three aquaria per treatment). Nubbins were regularly (5 times a week) moved between the three aquaria corresponding to each trophic condition to minimize any chances of aquarium or nubbin-pair effect. All aquaria were continuously supplied with filtered seawater pumped from 40 m depth, renewed at a rate of 12 L h^−1^. The temperature (26 °C ± 0.2 °C) was controlled by heaters connected to Elliwell PC 902/T controllers.

The first set of aquaria (heterotrophy) was kept in the dark for six weeks (until partial but not total bleaching was observed). In this condition, the corals did not receive any light but were fed 5 times a week at repletion with *Artemia salina* nauplii. Although they were bleached compared to their initial state, they still contained (photosynthetically inactive) symbionts in their tissue. The two other sets of aquaria (mixotrophy and autotrophy) received an irradiance of 200 ± 10 µmol photons m^−2^ s^−1^ (12 h:12 h light:dark) provided by 400 W metal halide lamps (HPIT. Signify France, Suresnes, France). Light was controlled by a LI-COR data logger (LI-1000) connected to a spherical quantum sensor (LI-193). While autotrophic corals were not fed with external food, mixotrophic corals were fed 5 times a week at repletion with *Artemia salina* nauplii. Corals were maintained in auto- or mixotrophy for 12 weeks.

After the incubation period, the corals were sampled and immediately flash frozen. Tissue was removed from the skeleton with an air pick and the slurry was homogenized with a Potter tissue grinder. The slurry was centrifuged three times at 3000× *g* for 10 min at 4 °C to separate the host tissue from the symbionts, according to [36]. The symbionts were further washed from the host debris by different centrifugations. The two fractions were then freeze-dried until subsequent analysis of their amino acid composition.

### 2.2. Compound-Specific Stable Isotope Analysis

Approximately 3 mg of lyophilized coral host tissue or symbiont was acid hydrolyzed in 0.5 mL of 6 nmol HCl at 150 °C for 75 min [37] under a nitrogen atmosphere inside a 4 mL glass vial with PTFE cap. The samples were cooled to room temperature and then HCl was evaporated under a gentle stream of nitrogen. The samples were neutralized twice with 0.5 mL ultra-pure water and evaporated with a gentle stream of nitrogen. We undertook chloroformate derivatization with the EZfaast amino acid analysis kit, with the slight modification of replacing reagent 6 with dichloromethane as a solvent. The EZfaast kit is considered to be the easiest, fastest, and safest method to work with and was already used successfully before [38]. For carbon analysis, we injected 1.5 µL in split mode (1:15) at 250 °C and we injected 2 µL in split mode (1:5) at 250 °C for nitrogen. Helium was used as a carrier gas at a constant flow of 1.5 mL/min. The amino acids were separated on a Zebron ZB-50 column (25 m, 0.25 mm, and 0.25 µm) in a Thermo Scientific Trace 1300 GC (Thermofisher, Waltham, MA USA). The gas chromatography (GC) conditions were set to optimized peak separation for the desired amino acids as follows: initial temperature 110 °C ramped to 240 °C at 8 °C per min and then ramped to 320 °C at 20 °C per min and held for 2.5 min. The separated amino acids were split on a MicroChannel device into two direction flows: one toward the Thermo Scientific ISQ quadruple for amino acid identification and the second toward the Thermo Scientific Delta V advantage for C and N isotope analysis. The ISQ condition was set to transfer line 310 °C, ion source 240 °C and scan range from 43 to 450 m/z mass range. To define the isotopic ratio of carbon and nitrogen, the separated amino acids were combusted in a Thermo scientific GC isolink II at 1000 °C for CO_2_ and N_2_. Before entering the Delta V for the N_2_ analysis the sample went through a liquid nitrogen cold trap to freeze down all other gases. From each sample, duplicates were injected for carbon and triplicates for nitrogen.

### 2.3. Data Analysis and Corrections

Stable isotope ratios were expressed in standard δ notation where the standard for carbon was Vienna PeeDee Belemnite (VPDB) and for nitrogen atmospheric N_2_ (Air). Separated amino acids were purchased from Sigma Aldrich and analyzed at the Geological Survey of Israel Elemental analyzer (1112 Flash EA, Thermo) interfaced with an isotope ratio mass spectrometer (IRMS, Delta V Plus, Thermo). To extend the nitrogen isotopic range, two certified amino acids (Alanine +43.25‰ and Valine +30.19‰) were purchased from Arndt Schimmelmann, (Indiana University). We used a standard that contains seven amino acids with the known isotopic ratio with isotopic range for nitrogen of −6.69‰ to +43.25‰. Since nitrogen is not added in the process of derivatization, corrections for nitrogen addition were not required. To account for the carbons that are incorporated during the derivatization process, the following correction factor for each amino acid was used:n_cd_δ^13^C_cd_ = n_c_δ^13^C_c_ + n_d_δ^13^C_dcorr_
where *n* is the number of moles of carbon, C_c_ the AA of interest, C_cd_ the derivatized compound and C_dcorr_ the empirically determined correction factor [39]. The standard AA was used to set C_dcorr_ for the later calculation of the isotopic ratio of our sample. It was injected three times after the combustion reactor oxidation for carbon and three more times for nitrogen to allow for drift correction, and it was injected again after a maximum of 18 injections. Since AAs differ in the presence of heteroatoms and functional groups, which may lead to different combustion efficiencies, an average of the standard injection from the beginning and the end of the sequence was used. For each sequence of nitrogen, a correction factor was applied based on the linear regression equation of the ratio between the known AA Isotopic ratio and the acquired result for the sequence. Since there is no addition of exogenous atoms of nitrogen in the derivatization process there is no need for correction per AA.

The Trophic level, or Trophic Position (TP) was calculated according to [38]. It measures the position of a species in a food web. Primary producers such as phytoplankton or plants (i.e., autotrophs) have a TP of 1, while the TP of primary consumers that eat primary producers (i.e., heterotrophs) is 2. The secondary consumers have a TP greater than 2. In mixotrophic organisms, such as corals, which live in symbiosis with photosynthetic dinoflagellates, a TP around 1 indicates that the association is mostly autotrophic (relying on the photosynthates acquired by the dinoflagellates), while a TP higher than 1 indicates an heterotrophic input [22,38].

The trophic position (TP) in this study is calculated according to the following equation:TP = ((δ^15^NGlu − δ^15^NPhe − β)/TDF_AA_) + 1.

The constant, β, is the difference between the δ^15^N values of glutamic-acid and phenylalanine AAs in primary producers (trophic position 1). The trophic discrimination factor (TDF_AA_) is the average δ^15^N enrichment relative to source AAs per trophic position. For this paper, we used the following constants β = −0.36; TDF_AA_ = 4.54 that are best fit to our derivatization method [38].

### 2.4. Statistical Analyses

All statistical analyses were performed using R v3.2.2 (R Development Core Team. 2015). For δ^15^N-AA data analysis: δ^15^N data were checked using Levene’s test (for the homogeneity of variances) and Bartlett’s test (for normal distribution) and failed to meet the assumptions for parametric tests. Therefore, δ^15^N data were tested using the Kruskal–Wallis test with the Wilcoxon rank sum test for the post-hoc analysis. All post-hoc *p*-values were adjusted using the Benjamini–Hochberg (BH) correction factor [40].

For δ^13^C-AA data analysis: δ^13^C data were normally distributed and were checked as described above. To test for differences between two treatments (autotrophy vs. heterotrophy), we used a non metric multidimensional scaling (nMDS) following a permutational multivariate analysis of variance (PERMANOVA) with distance matrices test [41] implemented as Adonis in the Vegan package. In Adonis, Euclidean distance matrix measurements were used as a response variable, considering the additive effect of the treatment (auto and hetero) and compartment (host and symbiont) factors [42].

To test for the effects of nutrition and compartment (host or symbiont) on the amino-acid isotope values, a two-way analysis of variance (ANOVA) was used. The data were checked using Levene’s and Bartlett tests to ensure they met the assumptions for parametric tests. The Tukey HSD was used as post hoc analysis for multiple comparisons between treatments with a significance level of *p* < 0.05.

## 3. Results

### 3.1. δ^15^N-AA

Due to their importance in determining trophic dynamics in animals [25,43], we focused our δ^15^N analysis on two amino acids, the trophic AA glutamic acid (Glu) and the source AA phenylalanine (Phe) (Figure 2). The δ^15^N-Glu and Phe values of the *Artemia* nauplii prey were 14.17‰ ± 0.22‰ and 8.03‰ ± 0.04‰, respectively. Feeding corals with *Artemia* induced a significant increase in the δ^15^N-Glu values (Kruskal–Wallis test, *p* = 0.00001) with significant differences between the three nutritional treatments (post-hoc Wilcoxon rank sum test, *p* < 0.005). The δ^15^N-Glu value of the mixotrophic (8.49‰ ± 1.98‰) and heterotrophic (12.65‰ ± 2.19‰) symbionts was significantly higher than the autotrophic group value (3.71‰ ± 1.23‰). The δ^15^N-Glu of the heterotrophic symbionts (12.65‰ ± 2.19‰) was closest to the value of the *Artemia* prey among all the treatments (Figure 2a). There was also a significant increase in the δ^15^N-Glu of the heterotrophic host (10.96‰ ± 1.45‰) compared to the autotrophic host (8.28‰ ± 1.62‰, post-hoc Wilcoxon test, BH, *p* < 0.04).

Phenylalanine values showed the same trend as the glutamic acid with a significant difference between nutritional treatments (Kruskal–Wallis test *p* = 0.002, Figure 2b). There was a significant increase (post-hoc Wilcoxon rank sum test, *p* = 0.001) in the δ^15^N-Phe value of the mixotrophic (7.60‰ ± 1.25) and heterotrophic (8.10‰ ± 1.82‰) symbiont groups compared to the autotrophic group (3.72‰ ± 1.21‰). On the contrary, the δ^15^N-Phe signature of the coral host did not vary between treatments (post-hoc Wilcoxon rank sum test BH *p* > 0.12). Finally, the δ^15^N-Glu and δ^15^N-Phe values were significantly different between the host and the symbionts in the autotrophic treatment (post-hoc Wilcoxon rank sum test, *p* = 0.001) but not in the other treatments.

The trophic index (TP) ratio (Figure 3) was different among nutritional treatments (Kruskal–Wallis rank sum test *p* < 0.001) but not between symbionts and host tissue. There was also no significant difference between the auto and mixotrophic groups (TP ratio between 1.13 and 1.40 ± 0.22). However, we did observe a significant increase in the TP ratio of the heterotrophic group, which was significantly different from the other groups (post-hoc Wilcoxon rank sum test, *p* < 0.007).

### 3.2. δ^13^C-AA

The δ^13^C-AA values for the nauplii ranged from approx. −20‰ for isoleucine and methionine to −26–30‰ for the other AAs (Table 1). δ^13^C-methionine of the *Artemia* was significantly higher compared to the δ^13^C-methionine of all symbionts and heterotrophic hosts (Figure 4, *p* < 0.05). In addition, the δ^13^C-phenylalanine of the *Artemia* was significantly lower than the δ^13^C- phenylalanine of all hosts and symbionts (Figure 4, *p* < 0.05). The δ^13^C values of the AA_ess_ tested for host and symbionts were not significantly different between treatments (Table 2), with the exceptions of methionine and isoleucine. The δ^13^C value of both host and symbiont methionine was 4‰ more negative under heterotrophy than autotrophy (Figure 4, *p* = 0.002). The δ^13^C of symbiont isoleucine was also ca. 2‰ more negative under heterotrophy than autotrophy (*p* = 0.046). The δ^13^C values of the mixotrophic group were very variable from one colony to the other, which contributed to the lack of difference between treatments. Therefore, the nMDS analysis was performed by taking into account only the autotrophic and heterotrophic feeding conditions (Figure 5). It showed a clear separation between the two treatments (PERMANOVA *p* = 0.002) but not between compartments (PERMANOVA *p* = 0.85) pointing to different carbon sources between treatments with similar carbon sources within the compartments.

## 4. Discussion

Corals have evolved to occupy some of the most oligotrophic waters, thanks to their symbiosis with dinoflagellates and tight nutrient recycling and exchange mechanisms within the holobiont. This pioneering study on the application of CSIA-AA to coral trophic ecology highlights the close nutritional cooperation between host and algal symbionts with regards to the heterotrophically-acquired nutrients. We observed, in both the mixotrophic and heterotrophic conditions, a significant shift in the δ^15^N signature of glutamic acid and phenylalanine of the symbionts compared to the autotrophic condition. The heterotrophically-induced δ^15^N shift in host tissue was significant in the heterotrophic but not in the mixotrophic condition. These results suggest that the δ^15^N enrichment seen within the AA, typically associated with fractionation due to heterotrophic feeding [25,43], is best represented in the symbionts rather than the host. This would suggest that the symbionts are the main sink for these trophic amino acids within the symbiotic association. In addition, using the δ^15^N_Glu_-δ^15^N_Phe_ to calculate trophic position [38], we found that heterotrophic treatments had a significantly higher TP than autotrophic and mixotrophic ones, without any significant difference between the two last treatments. Although it is expected that corals in the mixotrophic treatment were eating available *Artemia*, this did not occur in levels high enough to increase their TP significantly after several weeks. Finally, the δ^13^C values of host and symbionts AAs were similar, independent of the trophic condition considered, suggesting a metabolic continuum in this coral–symbiont carbon association. Overall, this study brings essential information on how CSIA-AAs can be used to trace auto- and heterotrophy in a coral–dinoflagellate symbiosis, information that will have to be taken into consideration for future studies using this technique to highlight the nutritional ecology of scleractinian corals.

### 4.1. Lessons from the Amino Acid Profile in Autotrophic Holobionts

Autotrophic holobionts, which do not have any external supply of particulate food, are thought to only be able to acquire carbon and nitrogen from the organic and inorganic sources dissolved in seawater or recycled from the host waste products [17,44]. The similarity in δ^13^C-AAs values between the coral host and the symbionts suggests a close relationship between the two partners and the use of the same carbon source for amino acid synthesis. In corals, symbionts indeed transfer most of their photosynthates to the host [36,45]. Then, the host uses this organic carbon source and its nitrogenous waste products to synthesize AAs such as serine and glycine [31,46].

Contrary to the δ^13^C-AAs values, the δ^15^N of glutamic acid and phenylalanine was different between the symbionts and the coral host (ca. 4‰ versus 8‰ respectively). Such differences cannot be due to a transamination process during AA transfer from symbionts to hosts, because in this case, the δ^15^N values of the symbionts would have been higher than those of the host [23,47]. By the same logic, one possibility would be that enriched host signatures were due to host transfer of AAs to the symbionts, leading to an enriched host signal. However, transamination for Phe is considered a minor pathway in healthy organisms, and it would therefore not result in large fractionations from the nitrogen source [23,48]. This difference in both Phe and Glu indicates that host-transferred AAs are unlikely the cause, since Phe would be expected to remain constant between the two compartments. The differences in δ^15^N between the two compartments are therefore indicative of either different nitrogen sources or biosynthetic pathways for the two AAs investigated [47].

Glutamic acid, a nonessential AA, can be synthesized both by the host and the symbionts (Figure 1). Since the two partners use glutamine synthase (GS)/glutamine oxoglutarate amino transferase (GOGAT) as the main pathway for ammonium incorporation into glutamic acid [31], the different δ^15^N values of glutamic acid between the partners are most likely due to the use of different nitrogen sources. In turn, phenylalanine is an essential AA and the host is thought not to be able to synthesize in metabolically significant levels [46], though it has been suggested that coral can produce some AA_ess_ (Figure 1). Here, the very close δ^13^C profiles of phenylalanine between host and symbionts associated with the different δ^15^N values suggest that phenylalanine is synthesized in both compartments from the same carbon skeleton but from two different nitrogen sources, as for glutamic acid.

The difference in the nitrogen signatures of glutamic acid and phenylalanine between the host and the symbionts could have several origins. If the host is indeed able to assimilate ammonium (NH_4_^+^) and dissolved organic nitrogen, only symbionts can use nitrate (NO_3_^−^) as a nitrogen source [49,50]. NO_3_^−^ can represent one third of the total nitrogen taken up by corals [11], and it can explain the different δ^15^N values of glutamic acid and phenylalanine in the host and symbionts of *S. pistillata*. In our culture system, corals continuously received a mean amount of 0.5 µM NO_3_^−^, which might explain the result observed. Additionally, the sources of ammonium can be different, as ammonium dissolved in seawater rather accumulates in the symbionts [51] whereas the host may rely more on ammonium derived from its waste products or transferred by its symbionts. Alternatively, the trophic increase in host tissue δ^15^N-phenylalanine/glutamic acid can be due to the digestion of supernumerary symbionts by the host, which is a way in which the holobiont recycles N within the system [52]. The above hypotheses, however, remain to be investigated.

### 4.2. Lessons from the Amino Acid Profile in Mixotrophic and Heterotrophic Holobionts

Coral–dinoflagellate symbioses are defined as mutualistic because both partners benefit from the association, mainly via exchanges of nutrients. Such benefits are clear for the coral host, which receives most of the symbionts’ photosynthates to cover its basal respiratory requirements [36]. In addition, the host receives a large proportion of other nutrients from the symbionts, such as phosphorus and, particularly, nitrogen [52,53]. The nutritional benefits of the symbiotic association for the algal symbionts are mainly the access to the waste products of the animal host [54] and enhanced photosynthesis due to the backscattering of light by the skeleton [55]. However, it has been suggested that, in hospite, symbionts have reduced growth compared to the free-living dinoflagellates due to the tight control of nutrient delivery from the host [54].

The results obtained here provide additional evidence of nutritional benefits for the algal symbionts. Our results indicate that symbionts acquire nutrients from one or all of the following pathways: (1) plankton-associated AAs transferred from the heterotrophic feeding of the host, (2) recycled nitrogen associated to these AAs, or (3) direct uptake of organic nitrogen prior to host access and recycling. We indeed observed a significant increase in the trophic position and in the δ^15^N signature (by 5 to 6‰) of glutamic acid and phenylalanine from autotrophic to mixotrophic/heterotrophic symbionts, as is generally the rule [27,56]. Heterotrophic AAs/nitrogen acquisition can be seen to be a benefit for the symbionts, which profit from the external feeding of the host. If the symbionts are indeed nitrogen limited in the host tissue, they can take advantage of such an additional nitrogen source. Once nitrogen becomes more available through heterotrophic feeding, this would incur symbiotic growth and division, which would in turn require more nitrogen uptake. This positive feedback loop of symbiotic nitrogen demand would therefore turn the symbiotic compartment into a nitrogen “sink”. The assimilation of heterotrophically-acquired food by symbionts has been highlighted previously in studies, but without identifying the type of compounds acquired [22,57,58]. In [22], the authors observed that the trophic index of the symbionts of several coral species covaried with that of their host, implying that AAs acquired through host heterotrophy are translocated to the symbionts.

Symbionts have two different ways of having access to the heterotrophically-acquired nitrogen, either by the direct assimilation of AAs or through internal N recycling. Dinoflagellates are indeed able to assimilate amino acids [58,59,60], and could have thus assimilated the ^15^N-enriched amino acids from the *Artemia*, after their digestion by the host. Another pathway could be through an increased availability in dissolved inorganic nitrogen through release of waste nitrogen by the host. It is likely that the host would assimilate and retain the lighter isotope due to kinetic fractionation, and release the heavier isotopes as waste products, available to the symbionts [52]. However, we suggest that the first pathway, the direct assimilation of prey-derived AAs by the symbionts, is the most likely because the δ^15^N-AAs of the symbionts almost reached the same value as the *Artemia*. In the case of nitrogen acquisition through food recycling by the host, the δ^15^N-AAs in the symbionts should have been higher than those of the *Artemia*.

Contrary to the symbionts, the mixotrophic host presented a smaller change in the δ^15^N value of glutamic acid with heterotrophy. This might suggest that this compartment does not keep a large proportion of this AA for its own use, and rather transfers it to the symbionts. The process here could be similar to that observed in the pea aphids-*Buchnera* symbiosis, in which aphids specifically exchange glutamic acid with *Buchnera* symbionts for other essential and nonessential amino acids [61,62]. In the heterotrophic treatment, however, the host exhibited a noticeable increase in its trophic index, suggesting that food is directly assimilated both in the host and the symbionts in such a situation. The observed increase indicates that the holobiont is not fully heterotrophic, since if this was the case, one would expect it to be more enriched than its *Artemia* food source. However, this is partially also due to the varying tissue turnover rates within the host and symbiont compared to our experiment length. Tissue turnover rates differ between compartments and amongst species, as well as between carbon and nitrogen pools, and have been shown to take hundreds of days for complete turnover [52,63]. Nevertheless, the increased δ^15^N-values of the coral holobiont between autotrophy and heterotrophy suggests that the coral holobiont is indeed reliant mostly on organic nitrogen.

The similar δ^13^C-AAs values between the coral host and the algal symbionts, independently of the trophic status of the symbiotic association, confirm that symbionts provide their host with carbon blocks or entire amino acids. Such a close relationship between δ^13^C-AAs of the host and symbionts has been highlighted in a previous study [21]. Symbionts, therefore, appear to be key partners for the provision of essential amino acids to the host, even when the latter is kept in darkness and the symbionts are photosynthetically inactive.

Overall, the δ^13^C-AAs values were variable between colonies maintained within the same trophic conditions. The fact that such variability was observed, even in the purely autotrophic condition, suggests that changes in δ^13^C-AA among coral colonies and feeding status cannot be easily interpreted. This may be due to a colony-dependent allocation of the autotrophically-acquired carbon between tissue growth, respiration or mucus release. Nevertheless, the δ^13^C-isoleucine of the symbionts presented a significant shift between autotrophy and heterotrophy, suggesting that the symbionts acquired this essential AAs from host feeding. Indeed, the δ^13^C-isoleucine of mixotrophic and heterotrophic symbionts (−21 to −23‰) drifted towards the *Artemia* value (−21‰) whereas the δ^13^C-isoleucine of autotrophic symbionts equaled −18‰. These results suggest a direct transfer of isoleucine to the symbionts, without deamination. Isoleucine is a branched amino acid and it has been shown that the microbial community associated with the coral *Porites astreoides* has three times the number of genes for the degradation of branched amino acids than for other amino acids [64]. This might also apply to *S. pistillata* symbionts, which can therefore be used as machinery to recycle branched amino acids obtained from the external food. In turn, the δ^13^C-methionine in the host tissue also shifted from autotrophy to heterotrophy, suggesting that the heterotrophic host acquired this AA from food, which is a key element in animal physiology, involved in epigenetic reactions and important for phenotypic plasticity [65]. The trophic index, although not significantly different between autotrophic and mixotrophic coral holobionts, continuously increases from the autotrophic to the mixotrophic and heterotrophic symbionts. It thus clearly shows a strong nutritional link between the host and the symbionts, as recently acknowledged [22] and emphasizes the importance of the symbionts in the acquisition and transformation of amino acids from the external diet and the translocation to the coral host.

Taken all together, the results of this controlled study show that the δ^15^N signatures of glutamic acid and phenylalanine of the dinoflagellate symbionts are an excellent proxy for heterotrophy. Therefore, this signal in symbionts can be used to trace heterotrophy in the field. It should be noted however, that the amplitude of the δ^15^N shift between autotrophic and mixotrophic holobionts is dependent on the length of the feeding, as well as the relative proportion of heterotrophic nutrient sources. Here, corals were continuously fed during several weeks, under conditions which allow adequate nutrient transfer time between host and symbionts, yet not enough time for complete tissue turnover [52,63]. Furthermore, the relative differences between host and symbiont enrichment could be due to the varying tissue turnover times between these two compartments, as well as varied nutrient recycling timing. Therefore, the application of these AA markers to in situ conditions should be valid in situations where corals are continuously exposed to high plankton concentrations. Our study also highlights similar δ^13^C values of essential amino acids between the host and symbionts, either suggesting that both partners use carbon compounds with a similar origin or that symbionts provide all AA_ess_ to the host even in the heterotrophic condition, when symbiont photosynthesis is inactive. Overall, differences in AAs biosynthesis and/or exchange rates with the host may partly explain the different susceptibility of corals to bleaching. For example, the capacity of *Porites* to biosynthesize cysteine without depending on its symbionts may account for its greater resilience to environmental stresses [33]. Additionally, in the giant sea anemone *Condylactis gigantea,* thermal stress resistance, as well as the incorporation of carbon into amino acids, was significantly lower in symbioses with algae of clade B than of clade A, suggesting that AA metabolism was indeed important in explaining the thermal stress resistance of each holobiont [66]. Overall, this study demonstrates that CSIA-AA has the potential to be used as a tool for studies regarding the nutritional exchanges within the coral–symbiodiniaceae symbiosis. However, further research is required to fully understand the fractionation pathways of both carbon and nitrogen during AA synthesis in the coral–dinoflagellate symbiosis. In addition, the effect of symbiont clade, host identity and environmental changes on this fractionation require more attention before using CSIA-AA to trace the trophic ecology of corals.

## Figures and Tables

**Figure 1 microorganisms-09-00182-f001:**
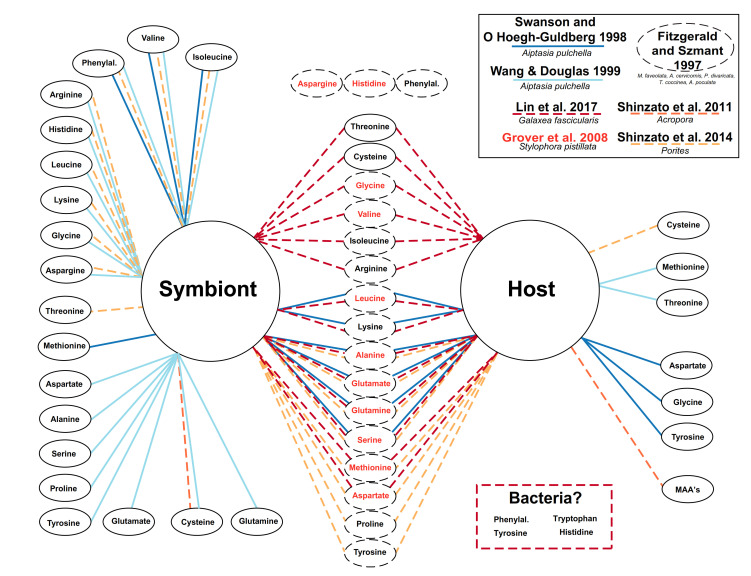
A schematic summarizing of previous research on amino acids (AA) and how they can be synthesized/obtained within the coral holobiont. Dashed lines are studies that used coral, and solid lines are those that used *Aiptasia*. Ellipses in the middle of the diagram are AA that can be synthesized/obtained by both host and algal symbionts: black hashed ellipses represent the pioneering work of Fitzgerald and Szmant; and red text indicates AA uptake through osmotrophy.

**Figure 2 microorganisms-09-00182-f002:**
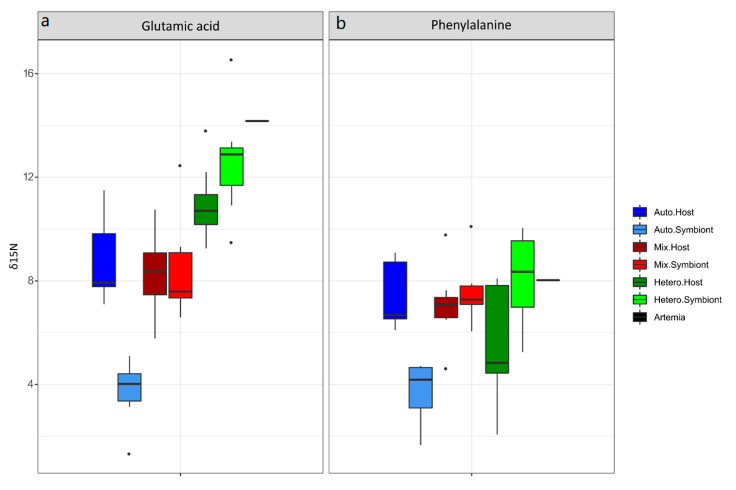
Changes in the δ^15^N value of the (**a**) glutamic acid and (**b**) phenylalanine according to the nutritional state of the coral holobiont: autotrophic host and symbionts are represented in blue, the mixotrophic host and symbionts in red and the heterotrophic host and symbionts in green.

**Figure 3 microorganisms-09-00182-f003:**
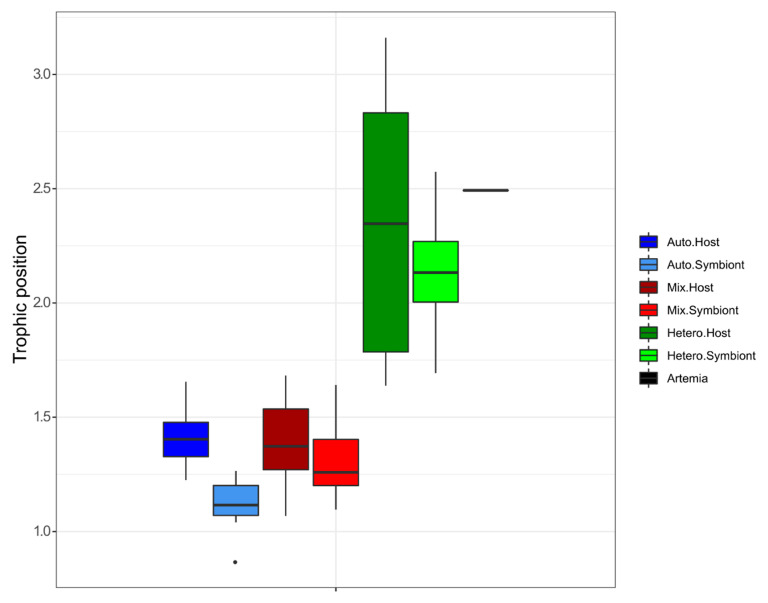
The value of the trophic position index according to the nutritional state of the coral holobiont: the autotrophic host and symbionts represented in blue, the mixotrophic host and symbionts in red and the heterotrophic host and symbionts in green.

**Figure 4 microorganisms-09-00182-f004:**
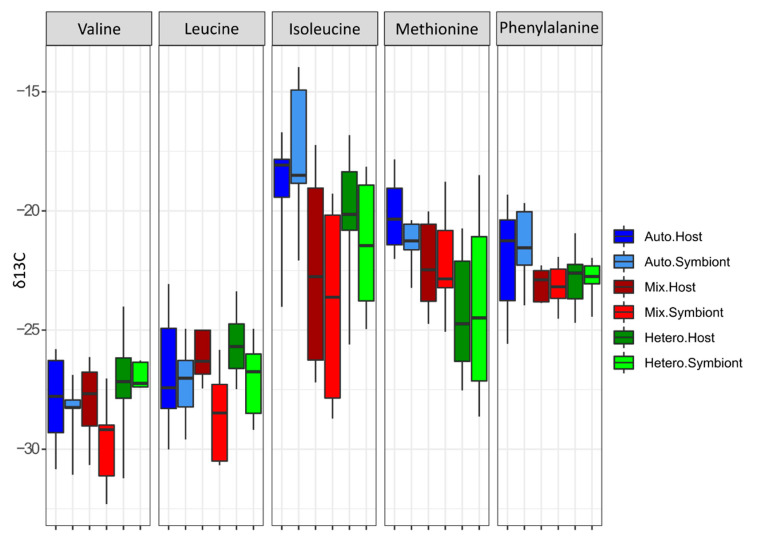
Changes in the δ^13^C value of the essential amino acids according to the nutritional state of the coral holobiont: the autotrophic host and symbionts are represented in blue, the mixotrophic host and symbionts in red and the heterotrophic host and symbionts in green.

**Figure 5 microorganisms-09-00182-f005:**
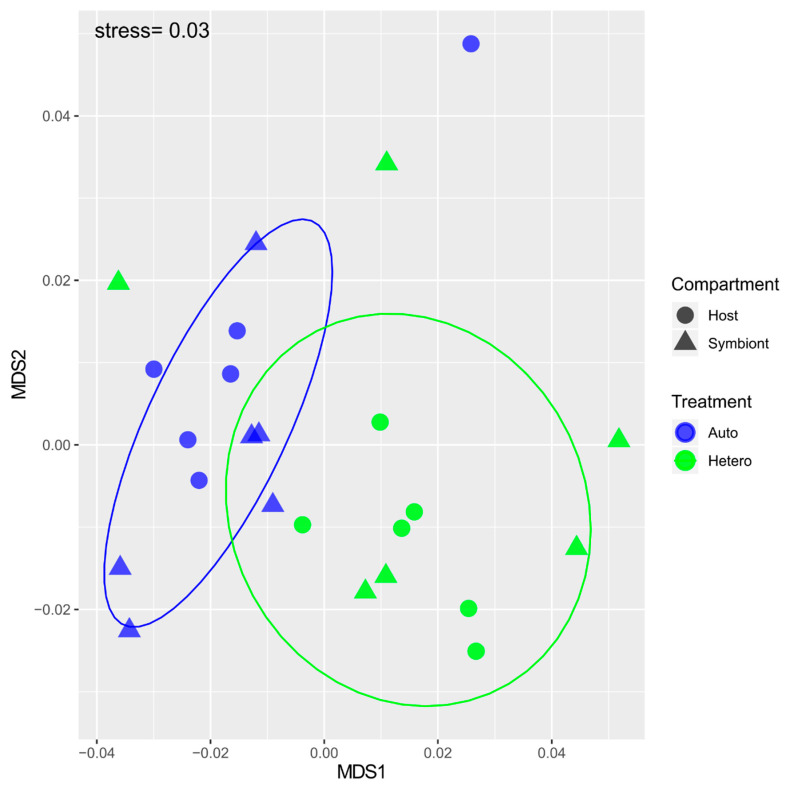
MDS performed with the δ^13^C value of the essential amino acids (Valine, Leucine, Isoleucine, Methionine, Phenylalanine) of autotrophic (blue symbols) and heterotrophic (green symbols) coral holobionts: host (circles) and symbionts (triangles). These data therefore include two nutritional treatments with two fractions, or 24 data points.

**Table 1 microorganisms-09-00182-t001:** The δ^13^C values of the essential amino acids contained in the *Artemia salina* prey (mean ± standard deviation).

Amino Acid	Mean δ^13^C	Standard Deviation
Valine	−30.527	1.393
Leucine	−27.465	1.064
Isoleucine	−21.086	1.046
Methionine	−19.899	1.021
Phenylalanine	−25.540	1.083

**Table 2 microorganisms-09-00182-t002:** The results from the statistical analysis testing the difference in δ^13^C value of the essential amino acids between the nutritional status, or the compartments (host versus symbiont). *p* ≤ 0.05 is significant.

Valine	Df	Sum Sqs	Mean Sqs	F. Model	*p*
Feeding	1	7.748	7.7482	2.84605	0.1081
Host.symbiont	1	0.074	0.074	0.02718	0.8694
Residuals	23	62.616	2.7224	0.88895	
Total	25	70.438	1		
**Leucine**					
Feeding	1	4.111	4.1106	1.2997	0.2648
Host.symbiont	1	5.182	5.1824	1.6386	0.215
Residuals	23	72.744	3.1628	0.88672	
Total	25	82.037	1		
**Isoleucine**					
Feeding	1	33.704	33.704	4.2303	0.050
Host.symbiont	1	0.118	0.118	0.0148	0.9034
Residuals	23	183.244	7.967	0.84419	
Total	25	217.065	1		
**Methionine**					
Feeding	1	76.003	76.003	12.9126	0.0014
Host.symbiont	1	1.164	1.164	0.1978	0.6554
Residuals	23	135.377	5.886	0.63694	
Total	25	212.545	1		
**Phenylalanine**					
Feeding	1	8.45	8.4502	3.6157	0.0705
Host.symbiont	1	0.194	0.1939	0.083	0.768
Residuals	23	53.752	2.3371	0.86146	
Total	25	62.397	1		

## Data Availability

All data are contained within the article.

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
