# Peer review of "Tracing the Trophic Plasticity of the Coral–Dinoflagellate Symbiosis Using Amino Acid Compound-Specific Stable Isotope Analysis"

_microorganisms, 2021, doi:10.3390/microorganisms9010182_

Round 1

Reviewer 1 Report

In the present study, the authors investigated the usefulness of tracking specific amino acids to study the nutritional exchange between coral host and symbionts and to trace coral heterotrophy, using holobionts with different nutritional modes (autotrophic/mixotrophic/heterotrophic). This is an interesting topic, as corals’ trophic plasticity in response to changing environmental conditions (e.g., thermal stress, eutrophication, etc…) has been identified as an important element in species’ resilience.

The topic follows the footsteps of two recent studies (Fox et al. 2019, Fujii et al. 2020), using compound-specific isotope analysis (CSIA) to assess the trophic plasticity in corals. These previous studies reported the usefulness of either d13C and d15N of amino acids to track the nutritional status of scleractinian corals. The novelty of the present work is showing that CSIA has the potential to be used also as a tool for studies regarding the nutritional exchanges within the Coral-Symbiodiniaceae symbiosis, something I feel should be made clearer. Considering that there are only the two aforementioned studies on a very similar topic, I wonder why these are just mentioned marginally in the Introduction and why they are not discussed in relation to the discoveries of this study. I think detailing the findings of Fox et al. and Fujii et al. a bit more and where applicable comparing them with those of the present study might help greatly to highlight the pioneering nature of the study (as the authors call it repeatedly).

SPECIFIC COMMENTS

- Correct “artemia” by “Artemia” in italic throughout the text. “nauplii” should not be in italic.

- Line 252: Capital letter for “Artemia”.

- Line 263: Correct to “Phenylalanine”.

- Line 293: Correct to “do not”.

- Line 388: Insert “,” after “association”.

- Line 390: Correct to “the latter”.

- Line 396: Delete “,” after “branched amino acids”.

- Line 403: Delete “themselves”.

Fox, M. D., Elliott Smith, E. A., Smith, J. E., & Newsome, S. D. (2019). Trophic plasticity in a common reef‐building coral: Insights from δ13C analysis of essential amino acids. Functional Ecology, 33(11), 2203-2214.

Fujii, T., Tanaka, Y., Maki, K., Saotome, N., Morimoto, N., Watanabe, A., & Miyajima, T. (2020). Organic carbon and nitrogen isoscapes of reef corals and algal symbionts: Relative influences of environmental gradients and heterotrophy. Microorganisms, 8(8), 1221.

Author Response

We would like to thank the reviewers for their constructive comments and suggestions, which have contributed to improve the manuscript substantially. Please find below the reviewer’s comments in regular font and our response in blue.

Reviewer 1

In the present study, the authors investigated the usefulness of tracking specific amino acids to study the nutritional exchange between coral host and symbionts and to trace coral heterotrophy, using holobionts with different nutritional modes (autotrophic/mixotrophic/heterotrophic). This is an interesting topic, as corals’ trophic plasticity in response to changing environmental conditions (e.g., thermal stress, eutrophication, etc…) has been identified as an important element in species’ resilience.

We thank the reviewer for his/her positive assessment of our work.

The topic follows the footsteps of two recent studies (Fox et al. 2019, Fujii et al. 2020), using compound-specific isotope analysis (CSIA) to assess the trophic plasticity in corals. These previous studies reported the usefulness of either d13C and d15N of amino acids to track the nutritional status of scleractinian corals. The novelty of the present work is showing that CSIA has the potential to be used also as a tool for studies regarding the nutritional exchanges within the Coral-Symbiodiniaceae symbiosis, something I feel should be made clearer.

We agree with the Reviewer that our study is novel in showing that CSIA can be used also as a tool for studies regarding the nutritional exchanges within the Coral-Symbiodiniaceae symbiosis. We have now highlighted this point in the abstract and in the conclusion of our study.

Considering that there are only the two aforementioned studies on a very similar topic, I wonder why these are just mentioned marginally in the Introduction and why they are not discussed in relation to the discoveries of this study. I think detailing the findings of Fox et al. and Fujii et al. a bit more and where applicable comparing them with those of the present study might help greatly to highlight the pioneering nature of the study (as the authors call it repeatedly).

In agreement with the Reviewer’s comment, we now present in the Introduction and discussion the main findings of Fox et al. and Fujii et al. We took into account the following message from both studies:

Fujii et al. have used the δ15NAA values of several coral species sampled in natural conditions to determine their trophic level (TL). TL of the examined coral samples ranged from 0.71 to 1.53, i.e., from purely autotrophic to partially heterotrophic. The TL of symbionts covaried with that of their hosts, implying that amino acids acquired through host heterotrophy are translocated to symbionts.

Fox et al. have used the δ13C of six essential amino acids (AAESS) of the coral Pocillopora meandrina. They found a complete separation in δ13C values of AAESS between the endosymbionts and the heterotrophic nutritional sources. In addition, they showed that AAESS δ13C values from 19 colonies of P. meandrina separated along a one‐dimensional continuum of autotrophic and heterotrophic nutrition.

SPECIFIC COMMENTS

- Correct “artemia” by “Artemia” in italic throughout the text. “nauplii” should not be in italic. Corrected throughout the text

- Line 252: Capital letter for “Artemia”. Corrected throughout the text

- Line 263: Correct to “Phenylalanine”. Corrected

- Line 293: Correct to “do not”. Corrected

- Line 388: Insert “,” after “association”. Corrected

- Line 390: Correct to “the latter”. Corrected

- Line 396: Delete “,” after “branched amino acids”. Corrected

- Line 403: Delete “themselves”. Corrected : waste products, available to the symbionts [53].

Reviewer 2 Report

General

Overall I think is a valuable and timely piece of work which advances understanding of compound specific stable isotope analysis in the coral-algal symbiosis. It is well written and thorough. Most of my suggestions are very minor and relate to small changes in language usage.

My main criticism of the piece of research would be the limited number of colonies used in the experiment. As pointed out in Fox et al 2019 Functional Ecology, there is large variation in the acquisition of AAess from autotrophic to heterotrophic sources between con-specific coral colonies at small scales in the wild.

Could this variation influence subsequent performance under laboratory conditions? Would the results of this study be robust to the inclusion of much higher numbers of colonies and the greater variation this would likely bring?

The statistical analysis and the bar plots per treatment are based on 6 replicates (but from 2 colonies)….which is really the very minimum….the study overall would be have been greatly improved if more colonies and  replicates were possible.

In addition….was the identity of the original host colony for each nubbin traced through the experiment? A test for the presence or absence of consistent differences between nubbins from each original colony would be very revealing and would support this study. If possible this could be included in supplementary results?

Abstract

Line 16. Suggest changing “of such a symbiosis” to “this symbiosis”

Line 21 to 22. The sentence beginning with “The trophic index” should be made clearer. “changes across conditions were more significant than in the host” should be clarified so that the reader can understand this independently.

Line 28. “A new tool to trace coral heterotrophy” is this statement consistent with Discussion lines 285 to 286 and the section starting line 409? Perhaps some qualification or greater detail is needed?

Introduction

I find the introduction in particular to be an excellent well written background to the subject, and good at highlighting the research gaps which currently exist in this subject.

Line 40. Change “mixotroph” to “mixotrophic”

Line 49. Remove “the” before predation

Line 70. change “indication on” to “indication of”

Line 81 AAess….to be consistent with earlier this should be AAess

Materials and Methods

Line 105. change to “with individuals of the scleractinian coral....” to ensure no suggestion this species is only found in the Red Sea.

Line 182 to 183. The equation here should really be written out fully in its own line (as was done in line 163 to 164 previously).

2.4 Statistical Analyses.

My main comment here would be consider splitting the statistics analysis section up according to the main datasets analysed….rather than having everything together in one section.

Line 190. This only mentions δ15N as being non-normally distributed. Were δ13C and AAs also tested in the same way and was the outcome the same?

Line 192. Suggesting including the reference to the BH correction factor in the bibliography.

Line 201. Relating to the previous point….is it therefore appropropriate to use Students t-test on this data?

Line 198. “All data were checked before to meet”. This needs changing to make sense. I also suggest being more specific about which datasets…..to make the whole statistics section easier to follow.

Results

Line 215. The p-value for the post hoc test here is given as 0.04. And this is significant…..was this after applying the Benjamini-Hochberg correction to the p-value? I would have expected this to be higher? Including a table of how the correction was applied to p-values as supplementary would be very useful.

Line 247. “δ13C values of the mixotrophic group were very variable from one colony to the other”. Do you mean colony here…..or nubbin? This relates to whether colony identity was tracked through the study.

Line 248 to 250. Suggest making it clear if the PERMANOVA test relates to a test on multiple amino acids, and making the terminology (group, compartment) match with the legend on the MDS plot.

Figure 5. There are more than 18 data points in this MDS. This must mean that the datapoints represent the δ13C of different essential amino acids? Is this correct? If so this should be made clear and also be shown in the legend.

Discussion.

The discussion is very thorough…..if anything it could possibly be condensed overall in each of the sections?

Line 271. suggest changing “a tight” to “tight”

Line 272. suggest changing “CSIA-AA on coral trophic ecology”  to “CSIA-AA to coral trophic ecology”

Line 350. Change to “as is generally the rule”

Line 360. The sentence starting “Dinoflagellates are indeed able to directly assimilate amino acids” doesn’t make sense to me. They can’t directly assimilate amino acids from artemia……it would still need to be via the host. It is still heterotrophically acquired. I know this is understood but perhaps this sentence (and some of this section) could be made clearer?

Author Response

We would like to thank the reviewers for their constructive comments and suggestions, which have contributed to improve the manuscript substantially. Please find below the reviewer’s comments in regular font and our response in blue.

Reviewer 2

Overall I think is a valuable and timely piece of work which advances understanding of compound specific stable isotope analysis in the coral-algal symbiosis. It is well written and thorough. Most of my suggestions are very minor and relate to small changes in language usage.

We thank the reviewer for his/her positive assessment of our work.

My main criticism of the piece of research would be the limited number of colonies used in the experiment.

We are sorry that this point was not made clear enough in the Material and Methods section. As described below with more details, we have used 6 genetically different colonies and not 2. Therefore, we believe that the number of colonies in this study was not limited and that our results represent the natural variability observed in the reef. Such variability can actually be observed with the d13C values of the amino acids which are quite variable from one colony to another one.

As pointed out in Fox et al 2019 Functional Ecology, there is large variation in the acquisition of AAess from autotrophic to heterotrophic sources between con-specific coral colonies at small scales in the wild. Could this variation influence subsequent performance under laboratory conditions? Would the results of this study be robust to the inclusion of much higher numbers of colonies and the greater variation this would likely bring?

We thank the reviewer for this very interesting question, showing that it is not so easy to interpret the d13C variations in corals.

As we have used 6 different colonies, we believe that our study presents robust results, particularly because they were obtained under laboratory-controlled conditions, where only the nutritional status changed, without any change in the other environmental parameters. This set up allowed us to assess the actual effect of feeding without any interaction with other parameters. Our study clearly shows that d13C-AA is very variable from one colony to another one, within the same trophic condition. Such variability, observed even in the purely autotrophic condition (no provision of heterotrophic food) suggests that variations in d13C among colonies and feeding conditions are difficult to interpret. This is maybe due to a colony-dependent allocation of the autotrophically-acquired carbon between tissue growth, or respiration and mucus release. Such difficulty in interpreting the changes in d13C signature has also been observed with bulk d13C signature of the coral tissue: a change by 1-2‰ in the bulk d13C with depth has been interpreted as heterotrophy in the seminal study of Muscatine et al. (1989), whereas it might also be due to a higher internal recycling of photosynthetically-acquired carbon with depth (as later suggested with several proxies by Einbinder et al. 2009). More controlled studies, with several coral species and trophic conditions, are thus needed for a good interpretation of the changes in d13C-AA in corals. On the contrary to the d13C-AA, d15N-AA is a very good proxy for heterotrophy, especially when measured in the symbionts. d15N-AA indeed showed more consistent values between colonies maintained under the same nutritional status, and shifts by several ‰ in symbionts from auto-to heterotrophy.

[Muscatine, L., Porter, J. W., & Kaplan, I. R. (1989). Resource partitioning by reef corals as determined from stable isotope composition. Marine Biology, 100(2), 185-193.]

[Einbinder, S., Mass, T., Brokovich, E., Dubinsky, Z., Erez, J., & Tchernov, D. (2009). Changes in morphology and diet of the coral Stylophora pistillata along a depth gradient. Marine Ecology Progress Series, 381, 167-174].  

The statistical analysis and the bar plots per treatment are based on 6 replicates (but from 2 colonies)….which is really the very minimum….the study overall would be have been greatly improved if more colonies and  replicates were possible.

We are sorry that this point was not made clear enough in the Material and Methods section. We used 6 and not 2 mother colonies and 1 replicate per condition of each of the mother colony. Therefore, as stated in the Material and Method section: “Three large nubbins were sampled from six mother colonies (n = 18). Nubbins were then identified, hung on nylon wires, and equally distributed in 3 different nutritional treatments (heterotrophy, autotrophy and mixotrophy), with 3 aquaria per treatment”.

As this sentence was not clear enough, we changed it into: “Three large nubbins were sampled from six genetically different mother colonies (n = 18). Nubbins were then identified, hung on nylon wires, and equally distributed in 3 different nutritional treatments (heterotrophy, autotrophy and mixotrophy), with 3 aquaria per treatment (so six nubbins per treatment from different mother colonies, distributed in three aquaria per treatment)”.

In addition….was the identity of the original host colony for each nubbin traced through the experiment? A test for the presence or absence of consistent differences between nubbins from each original colony would be very revealing and would support this study. If possible this could be included in supplementary results?

We agree that if only 2 colonies were used, this statistical treatment would have been needed. However, here, the six nubbins used per treatment are all different – as they come from 6 different colonies.

Abstract

Line 16. Suggest changing “of such a symbiosis” to “this symbiosis”: Changed accordingly

Line 21 to 22. The sentence beginning with “The trophic index” should be made clearer. “changes across conditions were more significant than in the host” should be clarified so that the reader can understand this independently. Changed accordingly

Line 28. “A new tool to trace coral heterotrophy” is this statement consistent with Discussion lines 285 to 286 and the section starting line 409? Perhaps some qualification or greater detail is needed?

Changed accordingly: “In addition, this study highlights δ15NAA as a useful tool for studies regarding the nutritional exchanges within the coral-symbiodiniaceae symbiosis”.

Introduction

I find the introduction in particular to be an excellent well written background to the subject, and good at highlighting the research gaps which currently exist in this subject.

We thank the reviewer for his/her very positive comment

Line 40. Change “mixotroph” to “mixotrophic” : Changed accordingly

Line 49. Remove “the” before predation : Changed accordingly

Line 70. change “indication on” to “indication of” : Changed accordingly

Line 81 AAess….to be consistent with earlier this should be AAess : Changed accordingly

Materials and Methods

Line 105. change to “with individuals of the scleractinian coral....” to ensure no suggestion this species is only found in the Red Sea. Changed accordingly

Line 182 to 183. The equation here should really be written out fully in its own line (as was done in line 163 to 164 previously). Changed accordingly

2.4 Statistical Analyses.

My main comment here would be consider splitting the statistics analysis section up according to the main datasets analysed….rather than having everything together in one section. Changed accordingly, we have rewritten this chapter.

Line 190. This only mentions δ15N as being non-normally distributed. Were δ13C and AAs also tested in the same way and was the outcome the same? δ15N data were not normally distributed whereas the distribution within the amino acids of the δ13C was normally distributed. This has now been made clearer in the manuscript.

Line 192. Suggesting including the reference to the BH correction factor in the bibliography. Done accordingly

Line 198. “All data were checked before to meet”. This needs changing to make sense. I also suggest being more specific about which datasets…..to make the whole statistics section easier to follow. Corrected and the sentence has been clarified in the revised manuscript.

Results

Line 215. The p-value for the post hoc test here is given as 0.04. And this is significant…..was this after applying the Benjamini-Hochberg correction to the p-value? I would have expected this to be higher? Including a table of how the correction was applied to p-values as supplementary would be very useful.

Yes, the post-hoc test of 0.04 is obtained after the BH (this point has been added to the manuscript). Before the correction the value was rather equal to 0.02.

Line 247. “δ13C values of the mixotrophic group were very variable from one colony to the other”. Do you mean colony here…..or nubbin? This relates to whether colony identity was tracked through the study.

As stated above, there was a mis-understanding in the Material and Methods section on the number of colonies and nubbins used in this study. As each of the six nubbins used per treatment corresponds to a different colony, the sentence above is correct.

Line 248 to 250. Suggest making it clear if the PERMANOVA test relates to a test on multiple amino acids, and making the terminology (group, compartment) match with the legend on the MDS plot.

In agreement with the reviewer’s comment, we changed the terminology from « groups » to « treatments » to fit the figure and the text

Figure 5. There are more than 18 data points in this MDS. This must mean that the data points represent the δ13C of different essential amino acids? Is this correct? If so this should be made clear and also be shown in the legend.

Yes, we have used all δ13C data. As we have for the MDS, two treatments (autotrophy and heterotrophy) and two fractions (host and symbionts), with six colonies for each condition and fraction, we have a total of 24 points on the MDS. We have changed the figure legend to make it clearer.

Discussion.

The discussion is very thorough…..if anything it could possibly be condensed overall in each of the sections?

Line 271. suggest changing “a tight” to “tight”: Changed accordingly

Line 272. suggest changing “CSIA-AA on coral trophic ecology”  to “CSIA-AA to coral trophic ecology”: Changed accordingly

Line 350. Change to “as is generally the rule”: Changed accordingly

Line 360. The sentence starting “Dinoflagellates are indeed able to directly assimilate amino acids” doesn’t make sense to me. They can’t directly assimilate amino acids from artemia……it would still need to be via the host. It is still heterotrophically acquired. I know this is understood but perhaps this sentence (and some of this section) could be made clearer?

We have changed this sentence accordingly: Dinoflagellates are indeed able to assimilate amino acids [59, 60], and could have thus assimilated the 15N-enriched amino acids from the Artemia, after their digestion by the host.

In addition, we have also clarified the following sentences: Another pathway can be through an increased availability in dissolved inorganic nitrogen through prey digestion and release of waste nitrogen by the host. It is likely that the host would assimilate and retain the lighter isotope due to kinetic fractionation, and release the heavier isotopes as waste products, available to the symbionts [53].